# A 2 μm Gallium Antimonide Semiconductor Laser Based on Slanted, Wedge-Shaped Microlens Fiber Coupling

Zhaohong Liu [1,2,*], Jiayue Wang [1], Ning Li [1,3], Zhongwei Yang [1], Shaowen Li [1], Sensen Li [1,2,3], Wei Wang [4], Heshig Bayan [4], Weining Cheng [3], Yu Zhang [2], Zhuokun Wu [3], Hongyu Sun [3], Yuanqing Xia [1], Yulei Wang [1] and Zhiwei Lu [1]

1    Hebei Key Laboratory of Advanced Laser Technology and Equipment, Tianjin 300401, China;
     202231903042@stu.hebut.edu.cn (J.W.); 202331905043@stu.hebut.edu.cn (N.L.);
     202221901035@stu.hebut.edu.cn (Z.Y.); 173757@stu.hebut.edu.cn (S.L.); sensli@163.com (S.L.);
     xiayq@hebut.edu.cn (Y.X.); wyl@hebut.edu.cn (Y.W.); zhiweilv@hebut.edu.cn (Z.L.)
2    Shanxi Key Laboratory of Advanced Semiconductor Optoelectronic Devices and Integrated System,
     Jincheng 048000, China; zhangyu@semi.ac.cn
3    China Electronics Technology Optoelectronic Research Institute, Tianjin 300308, China;
     202331905008@stu.hebut.edu.cn (Z.W.); 202331905065@stu.hebut.edu.cn (H.S.)
4    Changchun Institute of Optics, Fine Mechanics and Physics, Chinese Academy of Sciences,
     Changchun 130033, China; 202321901049@stu.hebut.edu.cn (W.W.); bayin888@sina.com (H.B.)
*    Correspondence: lzh@hebut.edu.cn

**Abstract:** Semiconductor lasers with a wavelength of 2 μm, composed of antimonide materials, find important applications in trace gas detection, laser medicine, and free-space optical communication, among others. In this paper, a more suitable microlens shape for 2 μm gallium antimonide semiconductor lasers is designed. Based on the fiber coupling efficiency model, the parameters of the designed slanting wedge-shaped microlens fiber are optimized to improve laser beam quality. The large tangent angle on both sides of the slanted, wedge-shaped microlens fiber is calculated using Snell's law, and the fiber core diameter and small wedge angle are determined through space fiber coupling experiments. After packaging the fiber coupling module with the chip, the laser output beam exhibits good overall symmetry in the spot with a uniform intensity distribution. The maximum output power is approximately 210 mW, demonstrating good power stability.

**Keywords:** antimonide semiconductor laser; 2 μm IR; fiber coupled; microlens

## 1. Introduction

Semiconductor lasers utilizing antimonide materials in the mid-infrared spectrum exhibit multiple gas molecule absorption peaks within the 2 μm band. Additionally, they exploit an atmospheric window characterized by high transmittance [1,2], rendering these lasers pivotal in the realms of trace gas detection, laser therapy, materials processing, and free-space optical communication [3–6]. Since the 19th century, extensive research efforts have been dedicated to mid-infrared semiconductor lasers, with a particular emphasis on antimonide-based semiconductor materials. These materials offer indispensable alloys, bandgaps, and bandstructures, constituting fundamental technologies in the contemporary landscape of mid-infrared semiconductor lasers [7–9]. Nevertheless, the current state of antimonide semiconductor lasers is encumbered by limitations in power output and beam quality, thereby constraining their potential applications.

In this paper, we employ the microlens fiber coupling technique to shape the laser beam, aiming to enhance coupling efficiency and improve beam quality. Based on different optical structures, coupling methods can be categorized into combination lens fiber coupling and microlens fiber coupling. Microlens fiber coupling involves the direct processing or attachment of a specific microlens to the end face of the fiber [10], enabling direct coupling with the semiconductor laser. The microlens shape can typically be conical [11],

hyperbolic [12], parabolic [13], cylindrical [14], and more. This paper utilizes the microlens fiber coupling method, directly coupling the microlens fiber with the light source. This approach offers several advantages, including a simple and compact structure, ease of alignment with the semiconductor laser chip, simple fabrication, and low production cost. A study on coupling a microlens fiber to a semiconductor laser shares similarities with the external combined lens fiber coupling scheme. The lens structure significantly influences whether high coupling efficiency can be achieved. Some representative work in related areas is presented below. In 2010, Yang et al. from Yishou University, Taiwan, compared the coupling efficiencies of tapered hyperbolic microlens fibers and tapered hemispherical microlens fibers and found that tapered hyperbolic microlens have much higher coupling efficiencies [15]. In 2014, Das from India studied the coupling efficiency of hyperbolic microlens fibers at two different wavelengths, 1.3 μm and 1.5 μm, using the ABCD matrix and obtained a higher coupling efficiency at 1.3 μm [12]. In 2015, Sanker at the Cape Institute of Technology, India, used a new type of fiber with an inverted-taper microlens at the fiber tip in order to maximize the coupling efficiency between a semiconductor laser and a single-mode fiber [16]. In addition, researchers have proposed a thermally expandable core fiber structure with a high numerical aperture to improve the coupling efficiency by using the thermally expandable core technique [17,18]. The thermal expansion core scheme offers higher coupling efficiency and greater lateral and longitudinal tolerances, but it has a lower tilt tolerance than conventional single-mode fiber schemes. In 2022, Nie et al. from the University of Wollongong, Australia, built a sensing system by coupling a semiconductor laser with optical feedback to a fiber [19]. The system can realize remote distance sensing, which is important for the practical application of semiconductor lasers coupled with optical fibers. At present, the development and application of microlens fiber coupling technology [20–23] are still hindered by challenges such as low coupling efficiency and a small mounting tolerance rate. These issues impede the effective resolution of problems related to the low power and poor beam quality of antimonide semiconductor lasers [24–28]. Therefore, it is necessary to analyze and study the microlens fiber end-face structure that is better suited for these lasers.

In this paper, we conducted a simulation study using ZEMAX OpticStudio2019 software to analyze the fiber coupling between common microlens fiber structures and antimonide semiconductor laser sources. The study aimed to compare coupling efficiency and mounting tolerance among different microlens fiber structures. Combining the beam characteristics of 2 μm antimonide semiconductor lasers with the advantages of wedge microlens and tapered microlens, we introduced a novel end-face structure named a slanted, wedge-shaped microlens fiber. This fiber demonstrated higher coupling efficiency, a larger modulation tolerance, and effectively improved the laser beam quality. The laser achieved a stable power output of 210 mW, with the output power showing increased stability near a current of 1.5 A. The RMS stability and peak-to-peak stability of the laser were 0.39% and 2.17%, respectively, during continuous output at a current of 1.5 A over two hours.

## 2. Theory and Simulation

The main common microlens fiber shapes include planar fibers, wedge microlens fibers, and tapered microlens fibers. Due to the large divergence angle of the semiconductor laser, direct coupling of a planar fiber to the light source results in poor coupling efficiency and low fiber mounting tolerance if the fiber is not processed. In this chapter, we simulate the coupling efficiency and mounting error rate of wedge and tapered microlens fibers. By combining the semiconductor laser chip used in this paper with the characteristics of wedge and tapered microlens fibers, we designed the slanted wedge microlens fiber. The structured fiber achieves a simulation coupling efficiency of 78.5% with uncoated end faces and significantly improved mounting tolerance.

In this paper, the M-1940-0500-A antimonide semiconductor laser chip, previously researched in the Superlattice Laboratory of the Institute of Semiconductor Research [29] at the Chinese Academy of Sciences, serves as a prototype design simulation light source. A

semiconductor laser light source is constructed based on chip parameters using ZEMAX sequence mode. Specific parameters of the optical fiber are configured in ZEMAX. The solid model of the fiber optic microlens is then established in SolidWorks 3D CAD2020 software. Subsequently, the non-sequential component in ZEMAX is utilized to import the microlens model, adjust its position, and nest it in the fiber optic end face to obtain the coupled model diagram of the microlens fiber. The geometric image analysis function is employed to calculate the coupling efficiency of the fiber, considering Fresnel reflection loss and optical polarization generated at the air-fiber split interface during the calculation process.

Typical angles for tapered microlens range from 45° to 100°. Microlens fibers with different taper angles can be selected based on the type of laser chip and desired light output performance. Taking the example of a 100° tapered microlens fiber, through simulation and analysis in ZEMAX, the coupling efficiency between the semiconductor laser and the tapered microlens fiber is determined to be 72.3%. Figure 1 displays the NSC solid model diagram of the optical fiber and illustrates the effects of positional and rotational degree errors on the fiber coupling efficiency. In Figure 1b, the mounting tolerances along the X and Y axes for the tapered microlens fiber are small, while the mounting tolerance along the Z-axis is relatively large. Figure 1c indicates that changes in the rotation angle of the fiber around its own axis do not affect the coupling efficiency of the tapered microlens fiber. This is primarily attributed to the circularly symmetric shape of the end face of the tapered microlens fiber, causing the rotation of the fiber around its own axis to have no impact on the light field pattern of the end face.

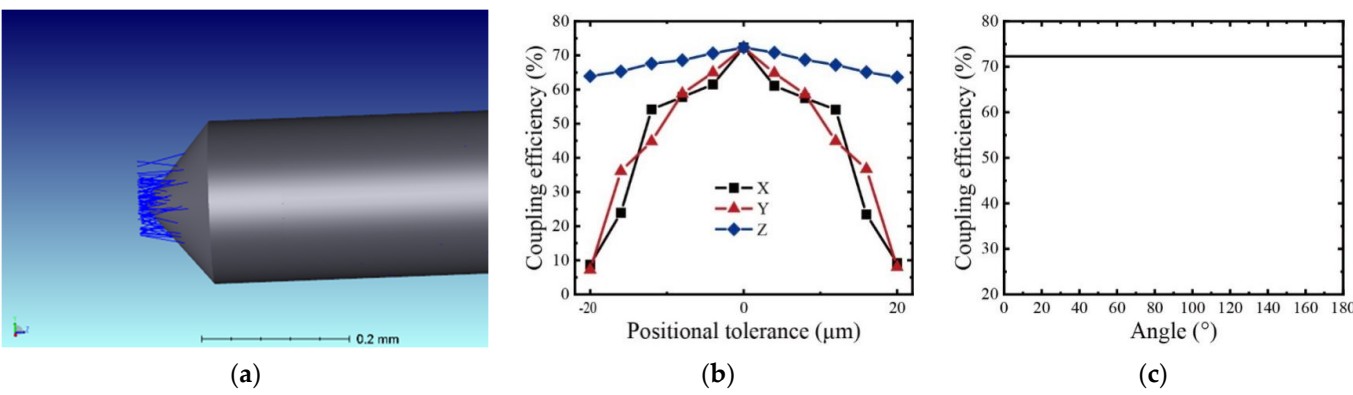

(**a**)      (**b**)      (**c**)

**Figure 1.** (**a**) Diagram of the NSC solid model of a tapered microlens fiber; (**b**) effect of tapered fiber position error on fiber coupling efficiency; and (**c**) effect of tapered fiber rotational degree error on fiber coupling efficiency.

For wedge-shaped microlens fibers, the size of the wedge angle has a crucial effect on the coupling efficiency of the fiber. We calculate the size of the wedge angle of the wedge microlens fiber according to Snell's law. Let the half wedge angle of the wedge lens be $\theta_x$ and the vertical divergence angle of the semiconductor laser chip be $\theta_\perp$, then we have the following relation:

$$\theta_\chi = \theta_\perp + \theta_1 \tag{1}$$

The following is known from Snell's law:

$$n_0 \sin \alpha = n_1 \sin \beta \tag{2}$$

Where $n_0$ is the refractive index of air; $n_1$ is the refractive index of the fiber core.

From practical application, it is necessary to make the incident light parallel to the z-axis direction, so that the following can be derived:

$$\sin \alpha = \sin\left(\frac{\pi}{2} - \theta_1\right) = \cos \theta_1 = n_1 \sin \beta = n_1 \sin\left(\frac{\pi}{2} - \theta_x\right) = n_1 \cos \theta_x \tag{3}$$

Bringing Equation (1) into Equation (3) yields the following:

$$\cos(\theta_x - \theta_\perp) = n_1 \cos\theta_x \tag{4}$$

The relation between the half-wedge angle $\theta_x$ of the wedge microlens fiber and the vertical divergence angle $\theta_\perp$ of the semiconductor laser is obtained after rationalization and simplification:

$$\theta_x = \tan^{-1}\left(\frac{n_1 - \cos\theta_\perp}{\sin\theta_\perp}\right) \tag{5}$$

The semiconductor laser chip utilized in this paper is known to have a vertical divergence angle of 50°, and the wedge angle of the wedge microlens fiber is determined to be approximately 70° according to Equation (5). Figure 2 illustrates the impact of position error and rotation error on fiber coupling efficiency for a 70° wedge microlens fiber. The highest coupling efficiency for the wedge microlens fiber, obtained through the optimization function of ZEMAX software, is recorded at 83.7%. In Figure 2b, the Y-axis tuning tolerance of the wedge microlens fiber demonstrates a significant improvement, and the Z-axis fiber coupling efficiency changes smoothly, while the X-axis tuning tolerance remains relatively small. In Figure 2c, it is evident that the rotation angle error of the wedge microlens fiber has a substantial impact on the coupling efficiency. This effect is attributed to the significant structural differences between the wedge microlens in the transverse and longitudinal axis directions. Any rotation introduces a mode-matching mismatch at the fiber end face, resulting in a pronounced cliff-type decrease in coupling efficiency.

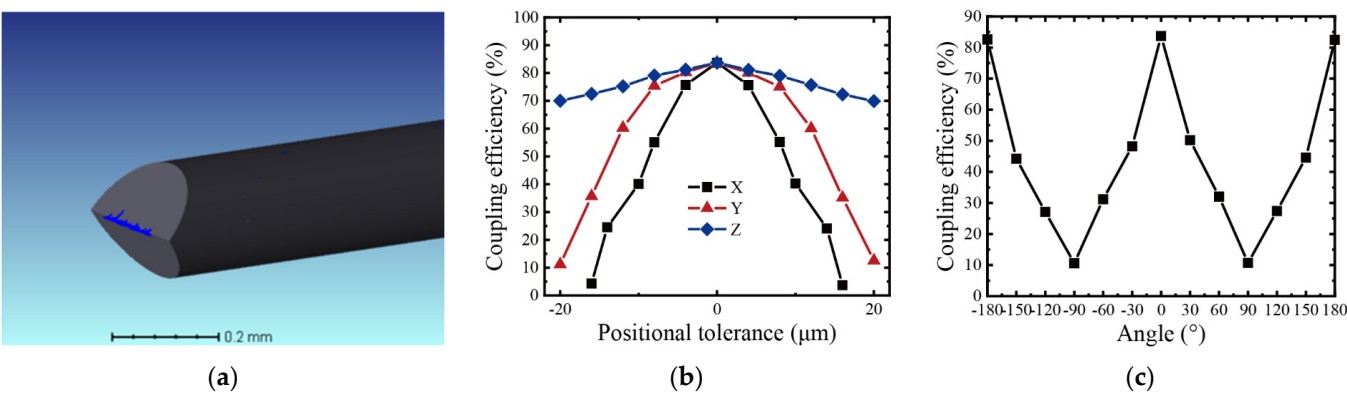

**Figure 2.** (**a**) Diagram of the NSC solid model of a wedge-shaped microlens fiber; (**b**) effect of wedge-shaped fiber position error on fiber coupling efficiency; and (**c**) effect of wedge-shaped fiber rotational degree error on fiber coupling efficiency.

Combining the findings from the above research and discussions, it is observed that the coupling efficiency is higher when compared with both the wedge microlens fiber and the tapered microlens fiber. However, the tuning tolerance of the wedge fiber is relatively small, especially when the fiber rotates around its own axis, leading to a significant impact on the coupling efficiency. This limitation is not conducive to practical experiments and tuning. On the other hand, although the tapered microlens fiber is not affected by the rotation angle, its coupling efficiency is lower. In response to these considerations, this paper proposes a new end-face structure called the slanted wedge microlens fiber, which is designed by combining the advantages of wedge microlens and conical microlens.

The end face of a slanted wedge microlens fiber is conceptualized as a slant cut on top of a wedge microlens, forming a small bevel to augment the numerical aperture of the fiber and enhance its light-harvesting capability. Consequently, the large tangent angle on both sides of the slanted, wedge-shaped microlens fiber is identical to the wedge angle of the wedge-shaped microlens fiber, measuring 70°. Utilizing the genetic algorithm optimization function in computer software, the small tangent angle of the slanted, wedge-shaped microlens fiber is optimized. Figure 3 illustrates the impact of changes in the small tangent

angle on fiber coupling efficiency, with a fixed large tangent angle of 70° on both sides. The optimization process reveals that the maximum fiber coupling efficiency occurs when the small tangent angle is approximately 28°. Through the multi-mode fiber coupling simulation function of ZEMAX, it is determined that the coupling efficiency of the slanted, wedge-shaped microlens fiber can reach 78.5% without coating at the end face, as analyzed through geometric image analysis in Optic Studio.

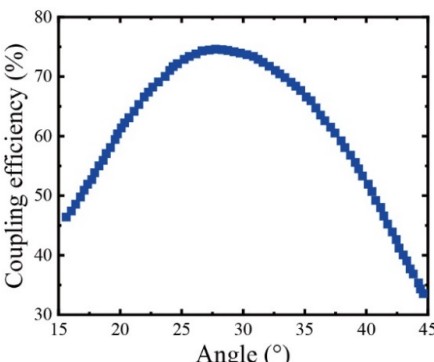

**Figure 3.** Fiber coupling efficiency versus small wedge angle of slanted wedge microlens fiber.

Figure 4 presents the NSC solid model diagram of the slanted wedge microlens fiber, along with a plot depicting the impact of position error and rotation degree error on fiber coupling efficiency. From Figure 4b, it is evident that the X-axis and Y-axis mounting tolerances of the optical fiber have been significantly enhanced, resulting in close adjustment tolerances in both transverse and longitudinal directions. The adjustment tolerance in the Z-axis direction is also improved compared to wedge microlens fibers, attributable to the increased numerical aperture that enhances light harvesting capability. In Figure 4c, the adjustment tolerance of the fiber in rotation around its own axis is improved compared to that of the wedge microlens fiber. When compared to the tapered microlens fiber, the adjustment tolerance between 0° and 40° is relatively large, and further improvement is needed in the adjustment tolerance between 60° and 90°.

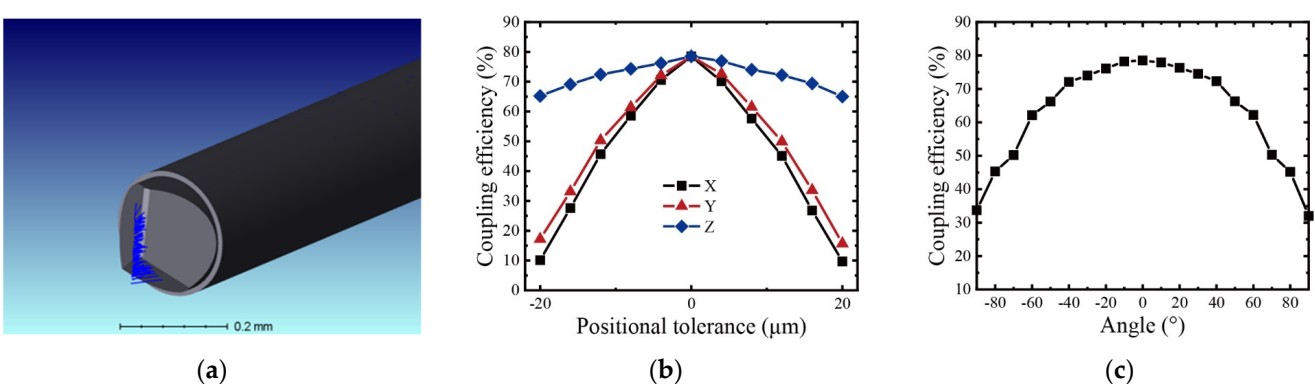

**Figure 4.** (**a**) Diagram of the NSC solid model of a slanted wedge microlens fiber; (**b**) effect of slanted wedge fiber position error on fiber coupling efficiency; and (**c**) effect of slanted wedge fiber rotational degree error on fiber coupling efficiency.

### 3. Experiment

*Optical System*

Firstly, we performed the spatial fiber coupling experiments on the slanted, wedge-shaped microlens fiber. Different small wedge angles of 105 μm and 200 μm of the slanted, wedge-shaped microlens fiber (no permeability-enhancing film on the microlens surface) were coupled to the semiconductor laser chip, respectively, and the large wedge angle on both sides of the slanted, wedge-shaped microlens fiber is unchanged at 70°, and the results are shown

in Figure 5. It can be seen from the figure that the coupling efficiency is 31.2%, 34.6%, and 29.3% for the small wedge angle of 20°, 30°, and 40° for 105 μm core diameter, respectively, and 51.2%, 52.5%, and 50.3% for the small wedge angle of 20°, 30°, and 40° for 200 μm core diameter, respectively. It can be seen that the coupling efficiency is higher for 105 μm and 200 μm core diameters when the small wedge angle of the slanted wedge microlens fiber is 30°. The coupling efficiency of the fiber is greatly improved when the core diameter is expanded from 105 μm to 200 μm, with 64.1%, 51.7%, and 71.7% improvement when the small wedge angle of the slanted wedge microlens fiber is 20°, 30°, and 40°, respectively. The above data were measured at a current of 1.5 A and a voltage of 1.85 V. From the experimental results, it is obvious that the 200 μm slanted wedge fiber with a small wedge angle of 30° has more advantages in coupling with the semiconductor laser chip.

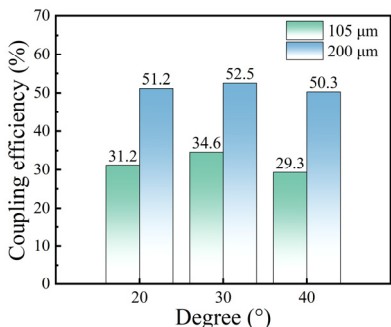

**Figure 5.** Experimental results of fiber coupling with slanted, wedge-shaped microlens with different small wedge angles of 105 μm and 200 μm.

Spatial fiber coupling experiments were performed using microlens fibers with wedge, tapered, and slanted wedge end faces. Figure 6a shows the physical and SEM images of the three microlens fibers. The coupling results of the three microlens fibers with a wedge angle of 70°, a taper angle of 80°, and a small wedge angle of 30° were put together for comparison, as shown in Figure 6b.

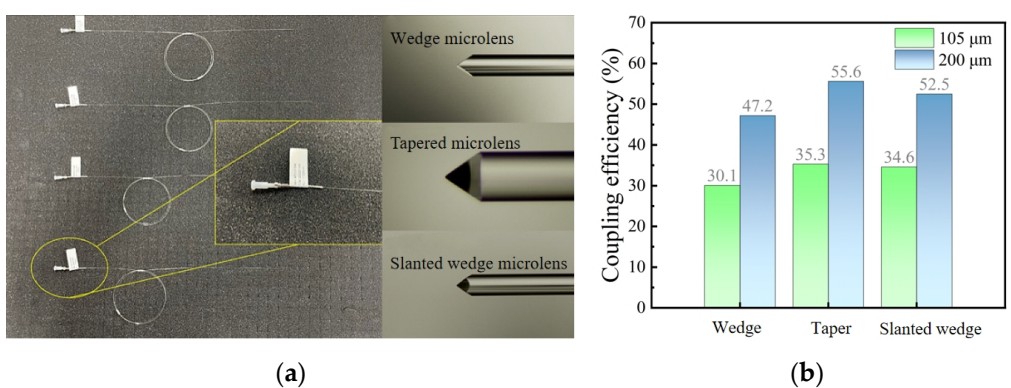

|     (**a**)     |     (**b**)     |

**Figure 6.** (**a**) Physical drawings of three optical fibers and microscopic microlens shapes of the fiber end faces; (**b**) comparison of coupling efficiencies of 105 μm and 200 μm wedge, taper, and slant-wedge end-face microlens fibers.

For the fiber coupling experiments, a 200/220 type numerical aperture of 0.22, a small wedge angle of 30° slanted, wedge-shaped microlens fiber, and a semiconductor laser chip are enclosed in a metal tube housing. The experiments involve fixing the fiber at the end to achieve maximum power, dispensing a fixed fiber, and finally sealing the setup to form a chip-fiber coupling module. Following the coupling between the semiconductor laser chip and the microlens fiber in the housing, the measured coupling efficiency exceeds 50%, meeting the anticipated standard.

Subsequently, after encapsulating the water-cooled module, the laser is assembled with the circuit module. The water-cooled module and circuitry are enclosed in the designed laser housing, secured with screws. Figure 7 illustrates the optical path of the gallium antimonide semiconductor laser. The core part of the semiconductor laser chip is the PN junction, which is dissected along the natural crystal surface to form two smooth end surfaces known as solvation surfaces. These surfaces function as planar reflectors, establishing a resonant cavity.

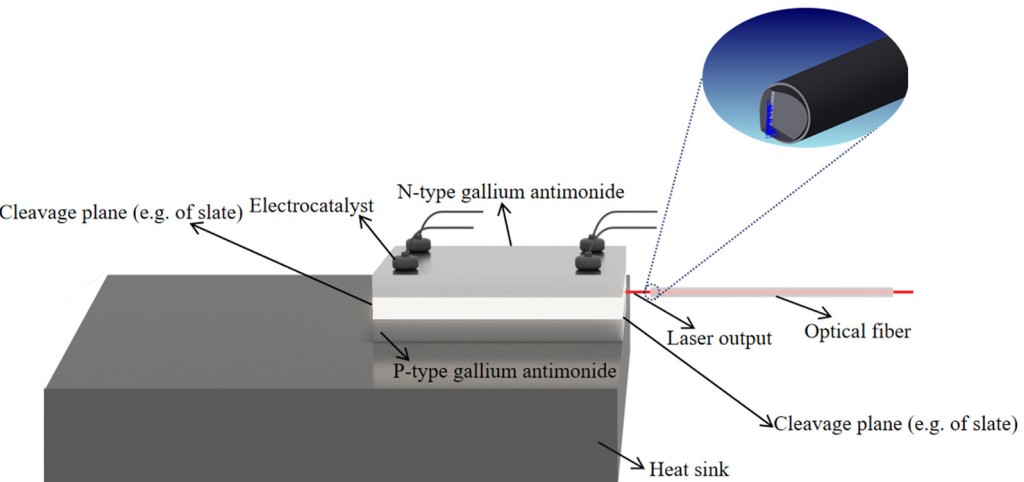

**Figure 7.** Gallium antimonide semiconductor laser optical path diagram.

Upon constructing the gallium antimonide semiconductor laser, a beam test is conducted to verify the anticipated results. In the performance test experiments, the focusing lens has a focal length of 500 mm. The beam quality analyzer is a CinCam product produced by CINOGY, Göttingen, Germany. The power meter head is a Vega handheld power meter manufactured by OPHIR, Israel, with the detector being a model 3A-P probe from the same company capable of detecting a power range of 15 µW–3 W. Additionally, the spectrometer used is a 771 series laser spectrum analyzer produced by British BRISTOL, featuring a spectral resolution as high as 2 GHz, a wavelength accuracy of ±0.0001 nm, and an optical rejection ratio exceeding 40 dB. The experimental test platform is completed by connecting the water cooler, gallium antimonide semiconductor laser, focusing lens, beam quality analyzer (power meter/spectrometer), and a laptop computer in the specified order.

## 4. Results and Discussion

Upon completing the construction of the platform, a comparison is made between the performance of the laser after fiber coupling and the performance of the chip when the light comes out directly. Figure 8 displays the PIV curves of the chip and the semiconductor laser.

In Figure 8a, the PIV plot of the chip reveals a threshold current of 0.3 A and a maximum output power of approximately 345 mW. The calculated average slope efficiency (W/A) for the chip, based on the measured data, is 0.17. It is observed that the power of the chip is more stable near the current of 1.5 A.

Figure 8b depicts the PIV curve of the gallium antimonide semiconductor laser, showing a threshold current of 0.2 A and a maximum output power of about 210 mW. The calculated average slope efficiency (W/A) for the semiconductor laser, based on the measured data, is 0.09. Similar to the chip, the power of the semiconductor laser is more stable in the vicinity of a current of 1.5 A.

A comparison with the chip test results reveals that the threshold current of the laser is reduced, attributed to the improved cooling effect of water cooling from an external water tank compared to natural convection cooling by air. This reduction in threshold current is a result of the enhanced cooling efficiency. Additionally, the power of the laser is found to

be diminished compared to the direct output power of the chip, owing to losses incurred during the fiber coupling and transmission process.

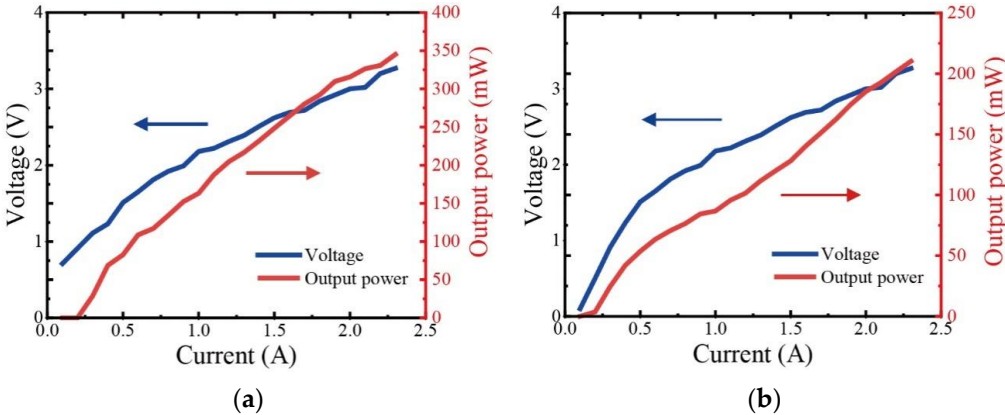

(a)

(b)

**Figure 8.** PIV curves of a chip and a semiconductor laser: (**a**) PIV curve of a semiconductor laser chip; (**b**) PIV curve of a gallium antimonide semiconductor laser.

Figure 9a displays the spectra of a gallium antimonide semiconductor laser at a current of 1.5 A. At this juncture, the primary wavelength of the semiconductor laser is observed to be 1935 nm, and the spectrogram exhibits a dendritic spectral distribution with a line width of approximately 15 nm. In Figure 9b, a comparison of spectrograms for gallium antimonide semiconductor lasers at different operating currents is presented. The spectral range of the semiconductor laser spans from 1910 nm to 1960 nm, with a gradual increase in current. Simultaneously, it is evident that the semiconductor laser output beam, under the influence of different currents, exhibits multiple longitudinal modes. This phenomenon is attributed to the laser chip for the FP cavity structure, where the frequency selection is not enough.

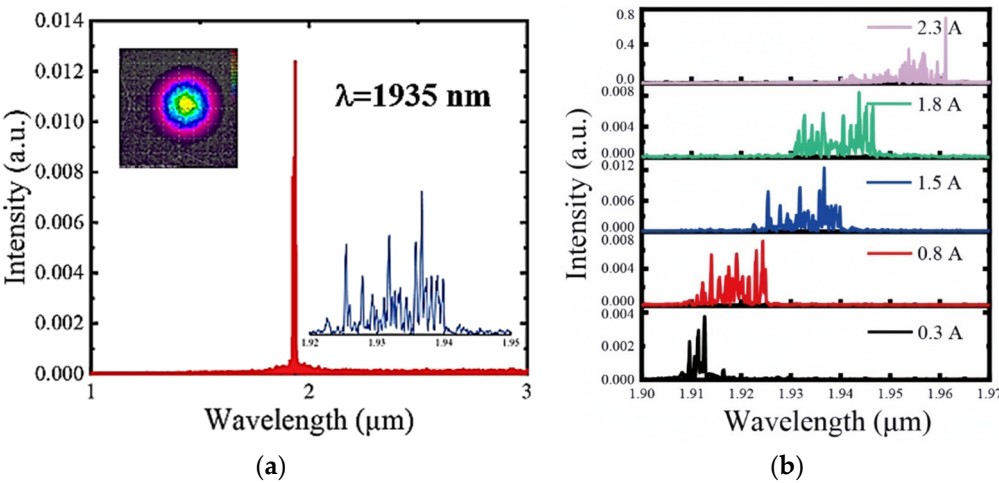

(a)

(b)

**Figure 9.** (**a**) Spectrogram of gallium antimonide semiconductor laser; (**b**) normalized spectra of gallium antimonide semiconductor laser at different operating currents.

The beam quality of the gallium antimonide semiconductor laser and the laser chip was subsequently tested using a beam quality analyzer produced by CINOGY. Figure 10 displays the output beam spot images of the chip and the gallium antimonide semiconductor laser at 1.5 A. In Figure 10a, the output beam spot of the chip is observed to be elliptical, with a relatively large spot length and width. The fast-axis and slow-axis divergence angles of the semiconductor chip are calculated to be 50° and 30°, respectively. Figure 10b illustrates the output beam spot of the laser, which is circular with good overall symmetry

and a Gaussian distribution featuring uniform intensity. The output beam spot of the laser is notably superior to that of the chip, indicating a significant improvement in beam quality.

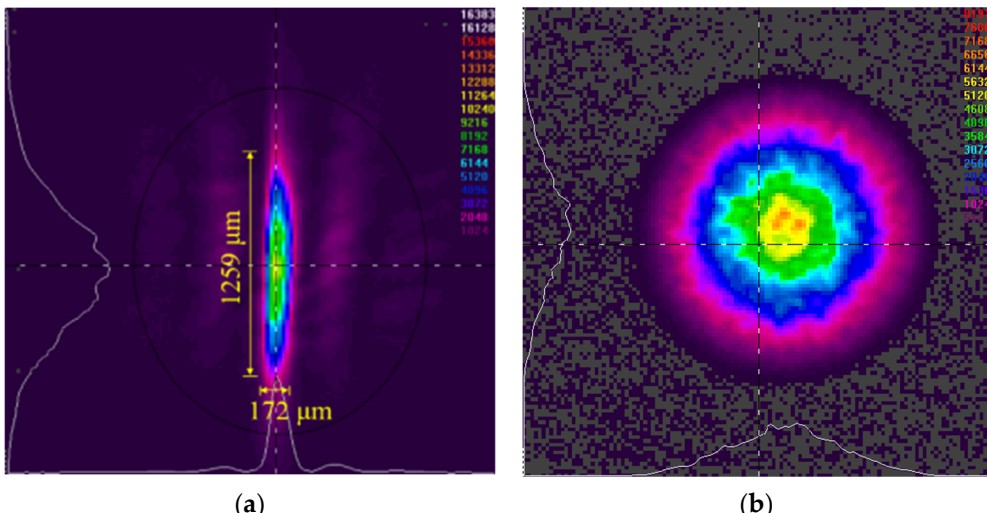

(a)  (b)

**Figure 10.** Output spot image: (**a**) output spot image of the chip; (**b**) output spot image of the gallium antimonide semiconductor laser.

For semiconductor lasers, the beam quality factor $M^2$ is ideally 1. In practice, the smaller the value, i.e., the closer to 1, the better the beam quality; conversely, the poorer the beam quality. Figure 11 shows the near-field spot images of a gallium antimonide semiconductor laser with an output from 100 mm to 500 mm at 1.5 A current measured using a beam quality analyzer, and the XY-axis spot size variation curve of the laser is obtained via data fitting. The output power of the laser is 128 mw at a current of 1.5 A. The beam quality factors in the XY direction are $MX^2 = 1.30$ and $MY^2 = 1.42$, and the divergence angles of the laser in the fast and slow axes are 35° and 28°, respectively, which shows that the beam quality of the semiconductor laser after fiber coupling has been greatly improved.

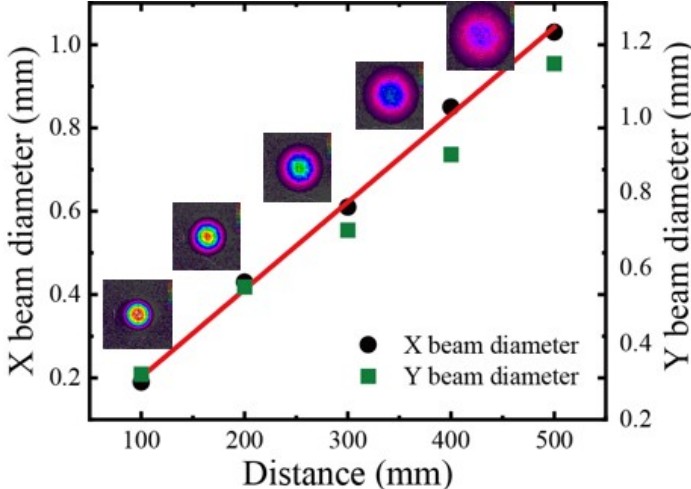

**Figure 11.** Relationship between output spot size and transmission distance of semiconductor lasers.

Figure 12 shows the corresponding power stability curve of the gallium antimonide semiconductor laser in the case of two hours of continuous output at a current of 1.5 A. Tested at an ambient temperature of 20° and an ambient humidity of 40% (no condensation). According to the results, the RMS stability and peak-to-peak stability of the gallium antimonide semiconductor laser can be calculated to be 0.39% and 2.17%, respectively.

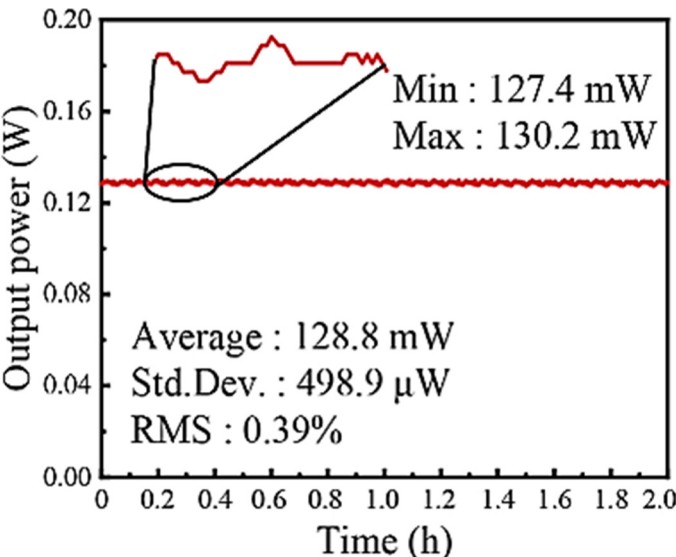

**Figure 12.** Power stability curve of gallium antimonide semiconductor laser.

## 5. Conclusions

In this paper, we calculated the angle of 70° for the large tangent angle on both sides of the slanted wedge microlens fiber according to Snell's law, and in the spatial fiber coupling of the slanted wedge microlens fiber, we found that the 200 μm slanted wedge fiber with a small wedge angle of 30° has obvious advantages when coupling with a semiconductor laser chip. We have used a 200/220 type optical fiber with a numerical aperture of 0.22 and a small wedge angle of 30° to complete the fiber coupling with a semiconductor laser chip in a metal tube housing. In this paper, we also completed the construction of a gallium antimonide semiconductor laser and performed a performance analysis of the laser. During the analysis of beam quality, we found that the diagonal wedge microlens fiber coupling effectively improved the beam quality of the laser. Analyzing the output power of the laser, we found that the threshold current of the semiconductor laser is 0.2 A and the maximum output power is about 210 mW. The RMS stability and peak-to-peak stability of this laser were 0.39% and 2.17%, respectively, under two hours of continuous output at 1.5 A current.

**Author Contributions:** Conceptualization, J.W.; methodology, J.W. and N.L.; software, Z.Y.; validation, S.L. (Shaowen Li); investigation, W.W.; resources, H.B.; data curation, W.C.; writing—original draft preparation, J.W.; writing—review and editing, Z.L. (Zhaohong Liu) and J.W.; visualization, Y.Z.; supervision, Z.L. (Zhaohong Liu), S.L. (Sensen Li), Z.W., H.S., Y.X., Y.W. and Z.L. (Zhiwei Lu); project administration, Z.L. (Zhaohong Liu); funding acquisition, Z.L. (Zhaohong Liu). All authors have read and agreed to the published version of the manuscript.

**Funding:** This research is funded by National Natural Science Foundation of China (61905064, 61975050), the China Postdoctoral Science Foundation (300428), the Hebei Province Postdoctoral special grant (B2022005003), the Research Projects of High Education Institutions of Hebei Province (QN2019201), the Natural Science Research Foundation of Hebei University of Technology (JBKYXX2002), the Open Project Program of Shanxi Key Laboratory of Advanced Semiconductor Optoelectronic Devices and Integrated (2022SZKF04), and the National Key R&D Program of China (2022YFB3606100, 9813E2301X04).

**Institutional Review Board Statement:** Not applicable.

**Informed Consent Statement:** Not applicable.

**Data Availability Statement:** Data are contained within the article.

**Conflicts of Interest:** The authors declare no conflicts of interest.

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
