# Peer review of "A 2 µm Gallium Antimonide Semiconductor Laser Based on Slanted, Wedge-Shaped Microlens Fiber Coupling"

_photonics, doi:10.3390/photonics11020108_

Round 1
Reviewer 1 Report
Comments and Suggestions for Authors
The authors have designed a slanted wedge-shaped microlens fiber to improve the coupling efficiency and beam quality of 2 2-um GaSb laser diode chip. The reviewer has the following comments:
1. The English language of the manuscript requires significant improvement. There are several places where words are repeated, figure numbers are referenced incorrectly, etc.
2. Introduction section: the first paragraph is on 2-um laser diode literature, which, the reviewer believes, is off track with the objective of the present study, which is on the design, development, and investigation of a new kind of slanted wedge microlens fiber for power coupling from the laser diode chip.
3. Introduction: the second paragraph discusses the standard type of fibers used for coupling power, but the literature review on the various new kinds of fiber tips for efficient power coupling from semiconductor laser chips, is somehow missing. The authors should include to enhance their introduction.
4. Section 2 needs a clear figure of the slanted wedge microlens with labeling of all angles and other parameters used in the analysis. It is very difficult to figure out from the text.
5. The last paragraph of page 3 needs to be revised as it is difficult to understand.
6. The authors should present the SEM and photographic images of their designed and manufactured fiber, which is missing.
7. The PIV results of the chip and fiber-coupled chip are compared. However, the authors should also compare with standard fibers (microlens fiber, standard wedge fiber, etc.) to show the effectiveness of their designed fiber in power coupling from the chip.
8. The emission spectrum of the laser is shown in Fig. 5(a). Does the red spike correspond to the lasing spectrum? If yes, then the inset spectrum of Fig. 5(a) does not show such a spike. Besides, such an emission spike is not visible in Fig. 5(b) as well with increasing current injection. The authors should clarify this in the text.
Comments on the Quality of English LanguageThe English language needs significant improvement.
Author Response
Dear Reviewer,
We would like to express our sincere appreciation to your warm work and thank you for this great opportunity for us to improve the manuscript “ 2μâ…¿ gallium antimonide semiconductor laser based on oblique wedge-shaped microlens fiber coupling, ID: photonics-2758844 ”.
We have carefully considered the comments from you, which help us to improve the manuscript substantially. Each suggested revision and comment brought forward by the reviewers was accurately incorporated and considered. We revised the content of the manuscript according to the valuable suggestions from you. We have also uploaded a response letter with point-to-point responses to your comments and highlighted specific responses in red font. And we have reworded all descriptions that were unclear. The changes in the manuscript are highlighted in blue, which shows in the revised manuscript.
Once again, thank you very much for your considerations of our manuscript.
Wish you all the best!

Reviewer 2 Report
Comments and Suggestions for Authors
1. In line 2, 2mm, m is symbol, not a letter.
2. In line33, line 37, line 38, line 43, these research findings should be cited as references.
3. In line 37, 2.1m? .
4. In line 89, 2-mm and 2mm words indicate inconsistency.
5. In line 167,the optical path of the GaAs laser is shown in Figure 3, not Figure 2, is gallium antimonide semiconductor laser ,not GaAs laser.
6. In line 171, line 182, line 238, GaAs semiconductor laser is gallium antimonide semiconductor laser.
7. In line 186, optical circuit diagram is optical path diagram.
Comments on the Quality of English Language1. In line 2, 2mm, m is symbol, not a letter.
2. In line33, line 37, line 38, line 43, these research findings should be cited as references.
3. In line 37, 2.1m? .
4. In line 89, 2-mm and 2mm words indicate inconsistency.
5. In line 167,the optical path of the GaAs laser is shown in Figure 3, not Figure 2, is gallium antimonide semiconductor laser ,not GaAs laser.
6. In line 171, line 182, line 238, GaAs semiconductor laser is gallium antimonide semiconductor laser.
7. In line 186, optical circuit diagram is optical path diagram.
Author Response

(The authors gave the same response as above.)

Reviewer 3 Report
Comments and Suggestions for Authors
Dear Authors,
Your results from calculations and experiments on fiber coupling optimization for a 2 micron gallium antimonide semiconductor laser are definitely worth to be published. But I think the quality of the presentation can be improved a lot specifically concerning the English language (see below).
Additionally I propose to improve the following issues:
1) It is not clear what the focus of the paper is: i) Is it the GaSb laser diode? Then simply to refer to a previous paper concerning the structure and fabrication of it would not be enough. ii) Is it the fiber coupling? Then it address a more general problem and the type and wavelength is not as important. Using this specific diode would then be only an example, what requires to restructure the paper, especially the introduction, what actually concentrates on GaSb lasers in general. iii) Or both? Then a comparison with other methods of fiber coupling like anti-reflection coatings is needed. You should make a decision here!
2) I suggest to provide a drawing in section 2 in order to make clear what angles you mean and what is the geometry considered. Formula (3) is trivial and can be omitted.
3) From Figure 4 I conclude that at currents of around 2 A the coupling efficiency is even higher than at 1.5 A. Why this point is not evaluated? How does this fit to the calculation?
4) Concerning the description of calculation: It is actually not clear what software is used for which calculation. Obviously three different products are mentioned: some genetic algorithm. ZEMAX, and OpticStudio.
5) The description of the setup (lines 172-184) need to be completely reworked. Please provide full sentences as this is much easier to read and understand.
6) Figure 4 is not only P-I but also U-I curve.
7) You might consider to evaluate the semiconductors red shift with current in Fig. 5.
8) Concerning the power stability in Figure 8: There seems to be a periodicity in the output what might be a hint of a periodic environmental influence (for instance temperature of the heat sink?). Can you comment o that? What was the current and temperature stability of the setup?
I have therefore voted to 'accept after major revision'.
Comments on the Quality of English LanguageI am wondering while 15 authors have read and agreed with the paper that the quality of the paper is that low. Some sentences are duplicated (lines 97-100, 175-177). Why the wedge angle is always a 'wedge angle angle'? Punctuation requires a space after a '.' or a ','. Sentences at lines 101-105 as well as 216-218 are completely not understandable. Someone should carefully read and edit the paper before submission!
Author Response

(The authors gave the same response as above.)

Round 2
Reviewer 1 Report
Comments and Suggestions for Authors
The authors have improved their manuscript and addressed the comments of the reviewer. However, looking at the revised manuscript and the comparison performed across different types of microfibers, the reviewer suggests the authors substantiate their manuscript by addressing the following comments:
1. The performance of the wedge (fig 2) and slanted wedge (fig 3) are comparable except in terms of tolerance. There is not much improvement?
2. Considering Fig. 6, again, the performance of the proposed microfiber is again comparable to the taper fiber and better than a wedge. There is not much improvement.
Comments on the Quality of English LanguageNone
Author Response
Dear Reviewer,
We would like to express our sincere appreciation to your warm work and thank you for this great opportunity for us to improve the manuscript “ 2μâ…¿ gallium antimonide semiconductor laser based on oblique wedge-shaped microlens fiber coupling, ID: photonics-2758844 ”.
We have carefully considered your comments, which have helped us to improve the manuscript substantially. Each of your suggestions and comments for changes have been accurately incorporated and considered. We have revised the content of the manuscript based on your valuable comments. We have also uploaded a response letter that provides a point-to-point response to your comments and highlights specific responses in red font.
Once again, thank you very much for your considerations of our manuscript.
Wish you all the best!
